# RefP2C: Reflective Paper-to-Code Development Enabled by Fine-Grained Verification

## Abstract

Reproducing machine learning papers is essential for scientific progress but remains challenging for both humans and automated agents. Analyses from prior studies reveal that the most prevalent issues arise during the code development phase, which is the foundational first step towards successful reproduction. Specifically, within this phase, agents often struggle to fully and accurately replicate implementation details such as mathematical formulas and algorithmic logic. Previous studies further show that reflection with explicit feedback improves agent performance. However, current paper reproduction methods fail to effectively adopt this strategy. This gap mainly arises from the diverse paper patterns, complex method modules, and varied configurations encountered in research papers. Motivated by how humans use systematic checklists to efficiently review complex code, we propose **RefP2C**, a **Ref**lective **P**aper-**to**-**C**ode Development framework that automatically extracts a paper's fingerprint-a comprehensive set of accurate and atomic criteria serving as high-quality supervisory signals. The framework first generates code based on the extracted information, and then leverages the fingerprint within iterative verification and refinement loop. This approach systematically detects discrepancies and produces targeted revisions to align generated code with the paper's specifications. Extensive experiments on the PaperBench Code-Dev benchmark have been conducted, RefP2C achieves 13.0% performance gap over baselines, and it correctly revises complex logical and mathematical criteria in reflecting, on which the effectiveness is obvious.

## 1 Introduction

Machine learning (ML) paper reproduction is the task of reproducing results of a paper without use of code from the paper's author (Semmelrock et al., 2025; Albertoni et al., 2023; Raff, 2019), which requires the reproducer or agent to develop and validate an implementation based on the paper's description[1]. With the rapid developments in Artificial Intelligence (AI), this task has become increasingly critical for accelerating scientific process and has already attracted attention from the community (Raff et al., 2025; Pineau et al., 2021).

Systematically evaluating this complicated task is hard due to the absence of unit tests, inherent code complexity and lack of reference code (Seo et al., 2025). Recent benchmarks like PaperBench (Starace et al., 2025) and ReproduceBench (Zhao et al., 2025) often adopt LLM-based progressive evaluation strategies. Their evaluation metrics primarily focus on three facets: code development, execution, and result match. However, evaluation results from PaperBench indicate that the performance of existing LLMs still lags significantly behind that of human experts, not to mention their limited performance on execution and result match (Starace et al., 2025). Although several methods (Seo et al., 2025; Zhao et al., 2025) have been proposed with the aim of achieving these goals, analyses of reproduction failures (Kon et al., 2025; Tang et al., 2025; Ni et al., 2025; Huang et al., 2024; Tang et al., 2024) reveal that *the most prevalent issues emerge during the implementation phase* (i.e., code development), such as missing essential implementation components. Since correct code is a prerequisite for any meaningful subsequent validation (Yan et al., 2025), ensuring its fidelity to the original paper is the most logical and essential first step towards successful automated paper reproduction, which is also the main focus of our work.

---

[1]Throughout our work, the term "paper reproduction" will refer specifically to this definition.

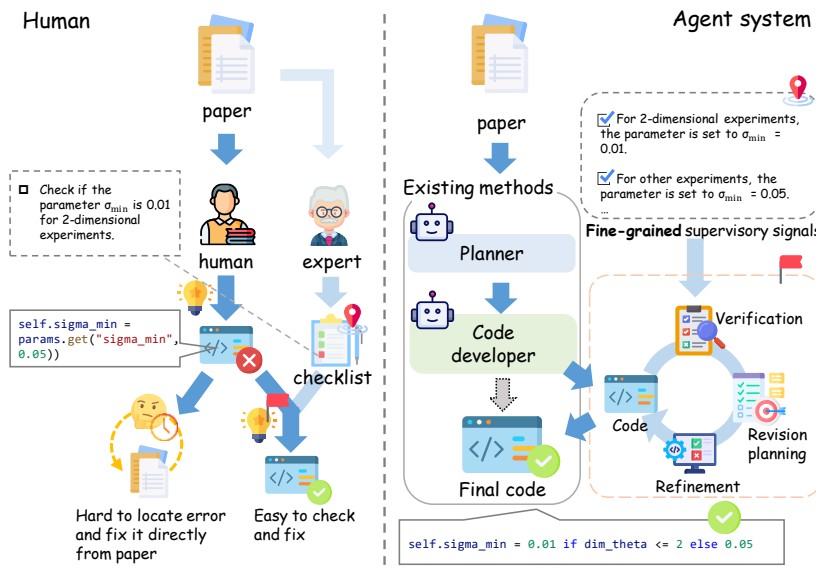

Figure 1: A comparison between human and agent system approaches to paper-to-code development. The left shows that manually reviewing code against a paper is labor-intensive, while a professional checklist can always offer a systematic way to assist. Our work (right) mimics the process by designing fine-grained supervisory signals, acting as a tailored checklist. This guides the agent through a reflective loop of verification and refinement to ensure the fidelity.

While PaperCoder (Seo et al., 2025) and AutoReproduce (Zhao et al., 2025) demonstrate that decomposition of code development is beneficial, they do not fully or accurately capture the exhaustive details specified in the paper, as empirically evaluated in PaperBench. This challenge primarily stems from the complexity of descriptions in papers, which also results in ambiguous supervisory signals for effective agent reflection (Seo et al., 2025; Wang et al., 2024a). High-quality supervisory signals can take the form of unambiguous scalars, such as unit test pass/fail results or compiler outputs (Jiang et al., 2025; Shinn et al., 2023; Hong et al., 2023). Motivated by these observations and the human practice of using systematic checklists to efficiently review code (see Figure 1) (Zhong et al., 2024; Amershi et al., 2019), a promising approach is to adopt similar supervisory signals to guide subsequent reflection (Shinn et al., 2023; Hong et al., 2023; Gu et al., 2024). However, how to obtain high-quality supervisory signals remains challenging due to the complexity of the papers.

In this paper, we address this challenge by introducing a novel **Ref**lective **P**aper-**to**-**C**ode Development framework, **RefP2C**. As shown in Figure 2, our approach first tackles the supervisory signal issue via automatic extraction of a paper's unique fingerprint—a comprehensive set of verifiable binary criteria encapsulating core implementation details. This fingerprint is generated by an automated agent workflow. During code development, we first construct a high-level framework and then fill in detailed implementations following the guides. In the refinement stage, a verifier reviews the code and provides pass/fail feedback, while a planner agent generates a revision plan as experience, executed by an editor agent through targeted minimal modifications. This framework enables reflective paper-to-code development guided by the fingerprints.

Extensive experiments on the PaperBench Code-Dev benchmark are conducted, and RefP2C achieves state-of-the-art performance, outperforming strong baselines by a significant margin of 13.0%. Its improvements come from correcting complex logical and mathematical errors when refining code. Our main contributions can be summarized as follows:

- Motivated by the checklist used by humans during code review, we design high-quality supervisory signals that can guide subsequent effective agent reflection.

- We propose RefP2C, a complete and reflective multi-agent framework that effectively integrates supervisory signals for paper's code implementation, evaluation, and revision.

- We show through extensive experiments that our paper-to-code development framework achieves state-of-the-art performance, and validate the effectiveness of our designed paper fingerprint.

## 2 RELATED WORK

### 2.1 AUTOMATED PAPER REPRODUCTION BY LLM

To accelerate machine learning progress, existing methods attempted to automate paper reproduction with LLMs (Starace et al., 2025; Seo et al., 2025; Zhao et al., 2025; Gandhi et al., 2025; Hua et al., 2025; Qian et al., 2024). To fully leverage the autonomous capabilities of LLM-based agent, PaperBench (Starace et al., 2025) proposes the BasicAgent built upon a ReAct framework, equipped with tools like a bash shell, Python executor to independently generate, run and submit code. IterativeAgent extends this by enabling iterative, step-by-step reproduction within a fixed time. Other recent methods design careful workflows involving paper analyzing, planning and coding. Paper-Coder (Seo et al., 2025) follows a single-pass workflow without evaluation with stages for planning implementation-level abstractions, analyzing modules, and generating code files while managing context for consistency. AutoReproduce (Zhao et al., 2025) employs collaborative research and code agents that gather external knowledge, generate data preprocessing scripts and iteratively refine implementations, primarily to ensure code executability.

Despite recent progress, existing methods still struggle to fully replicate the extensive detailed specifications in ML papers, which is the foundational step of paper reproduction. Many fail to capture subtle details or lack effective verification, resulting in incomplete development and limiting reliability. In contrast, our work directly confronts this problem by focusing on a systematic reflection process that ensures the fidelity of the first step.

### 2.2 REFLECTION AND VERIFICATION IN AGENTS

Reflection has emerged as the critical catalyst that turns static LLMs into reliable, self-improving agents across virtually every application domain (Renze & Guven, 2024; Shinn et al., 2023; Yao et al., 2023; Madaan et al., 2023; Ji et al., 2023; Jiang et al., 2024; Bo et al., 2024; Du et al., 2024b). These methods exhibit two key design questions: (i) what signal is used for critique, and (ii) how that signal is obtained. Firstly, evaluating code could use unambiguous scalar as a revision signal, derived from deterministic verifies like compiler diagnostics or unit-test pass-fail assessment (Shinn et al., 2023; Jiang et al., 2025; Hong et al., 2023; Huang et al., 2023). These signals provide objective binary or scalar indicators, but may not available for tasks which required human judge (Starace et al., 2025; Huang et al., 2023). Then, LLMs are widely adopted as a judge to provide detailed token-level critiques revision signal or high-level revision suggestions for code (Gu et al., 2024; Du et al., 2024a), which could be obtained relying on the reasoning ability of LLMs or combined with external tools like historical failure in memory or external knowledge base. Motivated by the unambiguous pass-fail unit-test signals, we extract atomic evaluation criteria to be developed, such that pass-or-fail binary assessment can be treated as the revision signal to guide the subsequent agent reflection.

## 3 METHOD

Existing methods fail to utilize effective supervisory signals for reflection. The inability to automatically generate clear and unambiguous pass-or-fail evaluation criteria significantly hinders reflection, leaving gaps in code alignment with the original paper. Motivated by how humans review code in software engineering, where tools like unit tests and checklists are used to ensure systematic and accurate verification, we propose RefP2C. When faced with difficulties in directly accomplishing paper-to-code development, RefP2C would typically prepare a checklist and apply effective agent reflection, simulating the reflective verification process in code review.

Specifically, it contains two stages:

- **Supervisory Signal Design**: As shown in Figure 2(a), the former is responsible for preparing effective supervisory signals. It is accomplished via an agent system involving multi-level paper guide extraction and grounding, standardization, and filtering, on which we obtain the comprehensive, accurate, and atomic evaluation criteria on details to be developed, which also are called fingerprint in this paper.

- **Reflective Code Development**: In the Stage 2, as shown in Figure 2(b), RefP2C leverages the extracted fingerprint to drive a complete code implementation and reflection workflow. It begins with an initial code implementation phase, which produces a base code, and then proceeds into an iterative multi-agent verification and refinement loop that systematically evaluates and revises the code to ensure alignment with the paper's specifications.

## 3.1 SUPERVISORY SIGNAL DESIGN

In practice, checklists are essential tools in ensuring that complex tasks, such as code review, are thoroughly and systematically performed. These checklists are designed to be both comprehensive and accurate, covering all relevant aspects of the task and ensuring that each item on the list is factually correct. Additionally, each item on the checklist typically represents an atomic unit. Drawing inspiration from this human approach, we establish two core principles for designing the paper fingerprint: i) **Comprehensive & Accurate:** the fingerprint must collectively cover all relevant details, while each individual criterion should be factually precise to prevent deviations from the original paper; ii) **Atomic:** each criterion

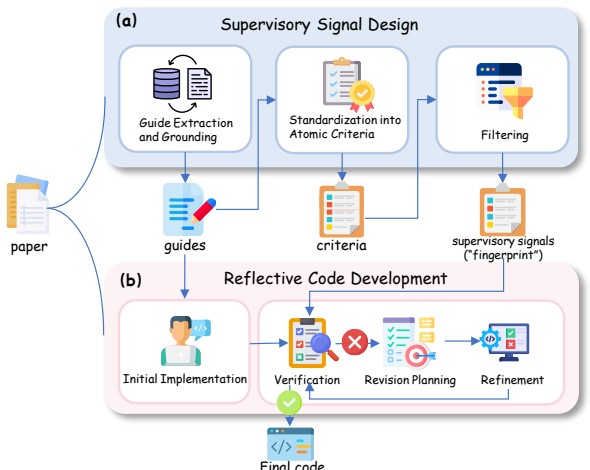

Figure 2: An overview of our proposed framework. The Supervisory Signal Design stage extracts fine-grained supervisory signals, which we call "fingerprint" for each paper. The fingerprint is then used in the Reflective Code Development stage to guide iterative verification and refinement.

should represent a single, verifiable unit that supports clear pass-or-fail binary evaluation. Guided by these principles, our agent system for extraction is as follows.

**Paper Guide Extraction and Grounding**  To achieve comprehensive and accurate evaluation criteria in the fingerprint, we first formulate three hierarchical guides to collect replication units at different levels, and then attach the corresponding paper sentence to each unit. This approach allows all potential evaluation points to be listed in the guides, with their source sentences prepared to facilitate evaluation by LLMs or human experts.

The process begins with the paper's Markdown content as raw input. The hierarchical extraction moves from coarse framework-level components, through detailed configuration units, to exhaustive paragraph-level scanning, progressively refining evaluation criteria from broad structures to detailed specifics to ensure comprehensiveness. Specifically, at the framework level, we aim to develop key components across all machine learning aspects—data, model, training, and evaluation. The agent system introduces each component unit by listing key sentences or paragraphs. At the configuration level, the agent system captures subtle implementation details and specific configurations in the paper, and preserves the configuration names alongside corresponding phrases or short sentences. Finally, to ensure fingerprint comprehensiveness, the agent system conducts an exhaustive scan for additional configuration units by extracting paragraph-by-paragraph, leveraging contextual memory to enhance understanding and capture any remaining sentences or equations.

To ensure the extraction is factually correct, the agent system links each unit in the guides to its corresponding paper sentences via an embedding retrieval strategy. Firstly, the paper is split into paragraphs and encoded with a sentence embedding model. For each extracted unit, the top-3 relevant paragraphs are retrieved. These paragraphs are then segmented into sentences, from which the agent will select the indices of multiple sentences that best correspond to the target unit. This action yields precise source references that facilitate reliable evaluation by LLMs and human experts.

**Standardization into Atomic Criteria**  As mentioned in Section 2.2, clear and verifiable criteria require atomicity, ensuring that each criterion can be evaluated with a simple pass-or-fail judgment. The agent extracts atomic evaluation criteria by decomposing and reformulating each unit into its

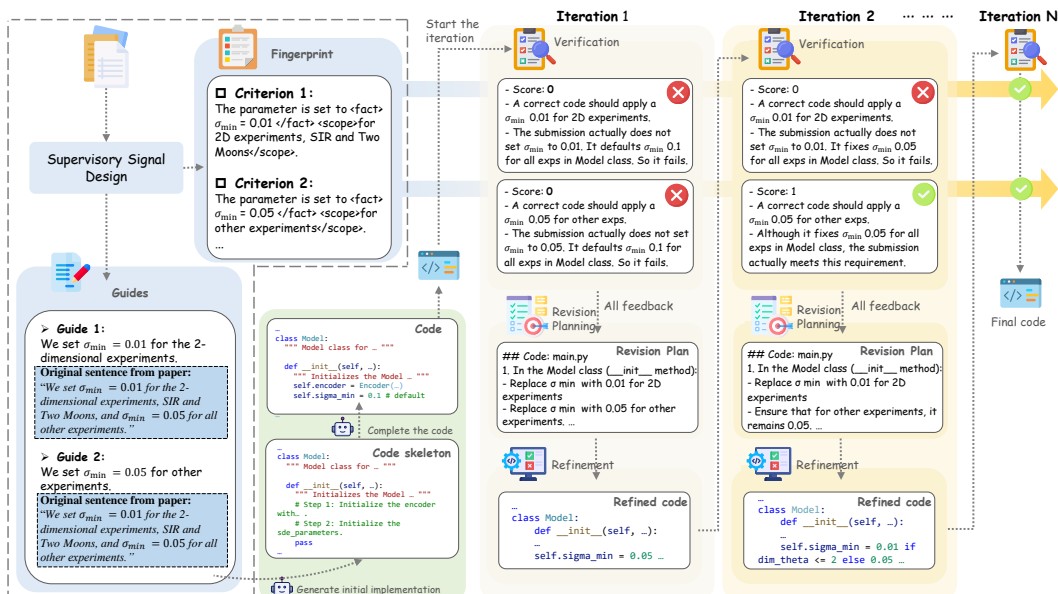

Figure 3: Details of our reflective code development. The dashed box highlights the "guides" and "fingerprint" generated in the first stage. In the second stage, a code agent uses these guides to create an initial implementation. This code is then iteratively verified against each criterion in the fingerprint. For any failed criteria, the agent analyzes the cause of failure, plans a correction, and revises the code accordingly. This iterative process concludes when the code passes all verifications or a maximum number of attempts is reached, yielding the final code.

atomic components, and then formulating them into *fact-scope* pattern. The "fact" could be hyper-parameters or implementation details, and the "scope" could be the dataset or task to which they apply. The agent then split each unit into multiple fact-scope pairs as independent criteria (see Figure 3 or Figure 10 for a detailed example). This structured evaluation criteria not only enables unambiguous verification but also greatly facilitates de-duplication by allowing direct semantic comparison of the <fact> components.

**Filtering** To ensure comprehensiveness, the numerous extracted criteria may contain repetitive and irrelevant items for paper-to-code development as shown in Appendix A.1. Therefore, the agent introduces a filtering stage to obtain the final paper fingerprint, which consists of clustering-based de-duplication and relevance-driven semantic filtering steps. For de-duplication, it first clusters the criteria based on <fact> embeddings, and then removes fact-scope pairs in each cluster that are identical in both fact and scope. Finally, it filters out the semantic-irrelevant or redundant criteria.

Finally, the comprehensive and accurate paper fingerprint is obtained, containing hundreds of atomic evaluation criteria. The detailed extraction steps and results are proposed in Appendix A.1. The fingerprint can be treated as a high-quality supervisory signal to guide simple pass-or-fail evaluation.

## 3.2 REFLECTIVE CODE DEVELOPMENT

To generate high-quality code aligned with the original paper, our framework develops code generation and refinement as a unified, reflective process. By integrating generation with iterative verification and refinement, this process equips the code agent system with self-reflective capabilities, enabling it to autonomously detect and correct errors, thus progressively improving fidelity of code development.

**Initial Implementation** During the stage, our code agent system leverages the raw, first two levels of multi-level guides extracted to produce the initial implementation, which serves as the base for subsequent reflection. These guides provide a more concrete and structured blueprint than the

detailed criteria, enabling the agent to generate code more logically without being constrained by fragmented details. The agent system first uses the high-level framework guides to construct a code skeleton, including essential classes and functions annotated with comments that outline the implementation steps. Subsequently, the agent populates this structured skeleton by systematically filling in methods and functions based on the extracted configuration guides. By doing this, the system obtains a code that mainly aligns with papers on framework and main functions. While the initial code captures core logic and modules, the code agent are not able to include all detailed implementations due to its ability.

**Verification**  To refine the initial code, a verifier first performs verification using the comprehensive fingerprint criteria (Section 3.1) as supervisory signals. Specifically, given a single criterion and generated code as input, the verifier first identifies the relevant portion of the code, then analyzes the expected and actual implementations. After comparison, it provides a pass/fail score and detailed textual feedback highlighting the expected criteria and observed discrepancies, as illustrated in Figure 3. The verifier transforms the trajectory (short-term memory) into internal comprehensive feedback to enable further self-reflection.

**Revision Planning**  Given the potentially large volume of feedback from assessing hundreds of criteria, immediate code refinement is impractical. Therefore, we introduce a revision planner that reflects all feedback collectively to develop a holistic understanding of the code's deficiencies. The planner then localizes issues within the code and synthesizes a comprehensive, step-by-step revision plan for each code file, serving as an experience (long-term memory).

**Refinement**  An editor is designed to refine the code based on the revision plan. Leveraging the plan, the full code, and the original paper as context, the editor performs targeted, minimal modifications in the order specified by the plan, refining the code file by file to ensure alignment with the planner. The refined code is then fed back into the verifier for subsequent iterations of evaluation and improvement, repeating this process until all criteria are satisfied or a predefined maximum number of iterations is reached. During this process, the results of each modification are stored as short-term memory, while the experience of each revision plan is recorded as long-term memory, both of which are fed back to the editor in subsequent iterations. Notably, the fingerprint serves as a form of static long-term memory, guiding the refinement to prevent deviation from the intended improvements.

In summary, leveraging the supervisory signals encoded in the fingerprint, the verifier generates detailed and actionable feedback on the code. The feedback is then integrated through a planner-guided revision process and executed by an editor, enabling systematic and targeted refinement. As a result, the generated code progressively aligns more closely with the original paper. More details are provided in Appendix A.2.

## 4 EXPERIMENTS

### 4.1 EXPERIMENTAL SETTING

#### 4.1.1 DATASET

We conduct experiments on the PaperBench Code-Dev benchmark (Starace et al., 2025) to evaluate our method. PaperBench assesses an agent's ability to fully and accurately reproduce a research paper and includes 20 Spotlight and Oral papers from ICML 2024. Each paper comes with a rubric manually co-developed with the original authors to ensure reliable assessment. The rubric is structured as a requirement tree, with leaf nodes specifying clear pass/fail criteria and parent nodes assigning weight scores. PaperBench's requirements are categorized into three types: code development, execution, and result match. Since our primary focus is on the first step of paper reproduction, we use a lightweight version of PaperBench ($66 per paper), PaperBench Code-Dev ($10 per paper), which only includes the code development requirements. We choose PaperBench over ReproduceBench (Zhao et al., 2025) because its human-verified rubrics and LLM-as-judge evaluation provide a more objective assessment, whereas ReproduceBench relies on Align-Score provided directly by an LLM.

### 4.1.2 EVALUATION METRIC

In the rubric, by grading all leaf nodes, i.e., assigning score 1 if pass and 0 otherwise, the parent node score is equal to the weighted average of their children's scores. The root-level score is the final replication score, which is denoted as root-level pass ratio $PR_{root}$ in our work. To directly measure the replication of each single criterion, we also report the leaf-level pass ratio $PR_{leaf} = \frac{\# \text{ passed leaves}}{\# \text{ of leaves}}$.

### 4.1.3 BASELINES

To make a fair comparison, we adopt four different agents designed fot this task. (a): **BasicA-gent** (Starace et al., 2025) is designed based on ReAct framework, and **IterativeAgent** (Starace et al., 2025) further extends with prompt engineering. We reuse the results reported. (b): **Paper-Coder** (Seo et al., 2025) completes the task through a forward-pass process of planning, analysis, and coding. We re-execute the official repository and report the results. (c): **AutoReproduce** (Zhao et al., 2025) reviews related papers and then retrieves knowledge and code to to inform its code generation process. We reuse the results reported. Moreover, we provide implementation details, including the specific LLMs, hyperparameters, and retriever models used in Appendix B.1.

## 4.2 PERFORMANCE COMPARISONS

Here, we compare the performance on PaperBench Code-Dev. For the leaf-level pass ratio $PR_{leaf}$, we only provide results for baselines where detailed scoring per paper is available. As shown in Table 1, it can be clearly observed that our method achieves the highest performance. The significant performance gap 13.0% and 17.5% compared with AutoReproduce and PaperCoder could demonstrate the effectiveness of our method, which owns to the reflective code development with fine-grained verifier.

Table 1: Performance comparison of different methods on the PaperBench Code-Dev benchmark. **Bold** numbers indicate the best performance in each metric.

| Method | LLM | $PR_{root}$ (%) | $PR_{leaf}$ (%) |
| --- | --- | --- | --- |
| BasicAgent | o3-mini-high | 6.4 | — |
| IterativeAgent | o3-mini-high | 17.3 | — |
| PaperCoder | o3-mini-high | 45.1 | 41.2 |
| AutoReproduce | o3-mini-high | 49.6 | — |
| RefP2C | o3-mini-high | **62.6** | **61.0** |

Furthermore, looking deeper into the leaf-level criteria, our method achieves an 19.8% higher $PR_{leaf}$ compared to the baseline PaperCoder. On average, our framework replicates 30 more requirements per paper than PaperCoder. Moreover, as shown by the paper-level performance gains in Figure 4, our method passes more evaluation criteria in 15 out of 20 papers, especially on tasks requiring high mathematical fidelity (e.g., `mechanistic-understanding`, +52.8%) and complex algorithmic logic (e.g., `pinn`, +65.1%). This demonstrates that our framework excels at capturing and faithfully developing the critical implementation details of a paper. For the 4 papers with decreased performance, the observed performance drop is mainly attributable to the discordance between semantics and syntax within the generated code (Gao et al., 2023; Wang et al., 2025; 2024b). For example, in paper `lca-on-the-line`, the generated code `model.feature_extractor()` is semantically correct and aligned with the evaluation criteria "load frozen pretrained features". However, it fails the LLM-as-judge, which indicates that the correct implementation should be `model.fc()`. Such semantic–syntactic mismatches partly account for the performance drop on PaperBench Code-Dev and would need to be addressed at the execution level. A detailed analysis is provided in Appendix B.2.

## 4.3 CASE STUDY

To further clarify our framework's effectiveness, we then conduct a case study. We select the `mechanistic-understanding` paper for this analysis as it not only represents the significant performance gain of our method over PaperCoder (+52.8% $PR_{leaf}$) but also has a concise rubric (36

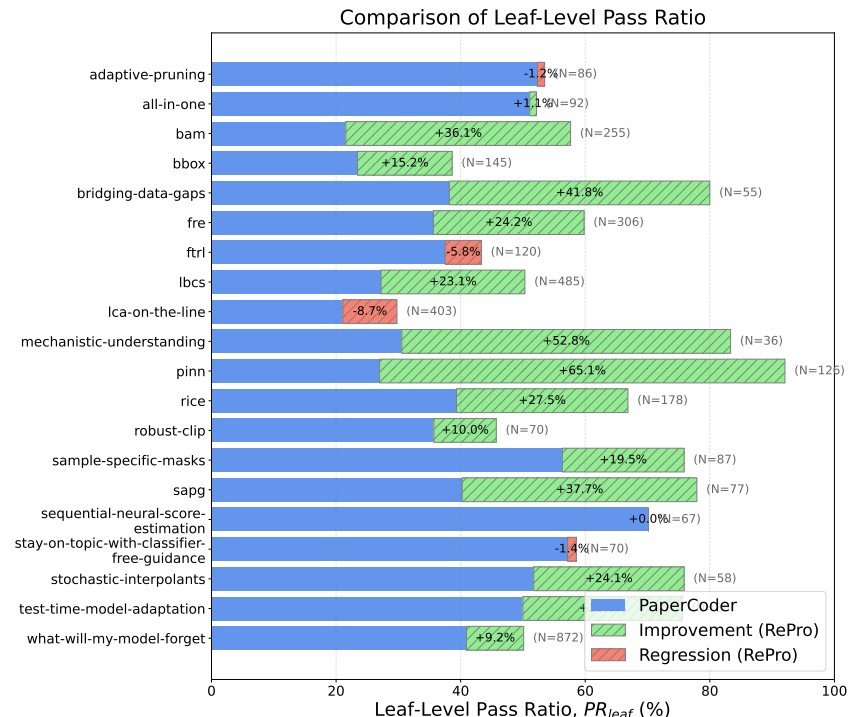

Figure 4: Per-paper comparison of leaf-level pass ratio (%). The blue bars show the baseline performance of PaperCoder. The green (improvement) or red (regression) hatched bars show the change brought by our RefP2C framework. The total number of rubric requirements for each paper's code development is noted on the right.

evaluation criteria), allowing for a focused analysis. On this paper, our method successfully passed 30 criteria (83.3%), while PaperCoder passed only 11 (30.5%), achieving 19 more correctly implemented requirements with no regressions. As highlighted in Table 2, the improvements achieved by our framework are primarily concentrated in two categories: mathematical fidelity and algorithmic logic. The first category, mathematical fidelity, pertains to the accurate implementation of detailed mathematical operations, such as matrix decompositions and vector transformations. The second category, algorithmic logic, involves the faithful development of intricate procedural steps such as selection, ranking, and thresholding mechanisms. The substantial gains achieved by our method in these areas demonstrate its superior capability to precisely interpret and implement complex mathematical constructs and algorithmic procedures. More details can be found in Appendix B.3.

Table 2: Categorization of implementation improvements in our code development compared to PaperCoder, with examples from rubric in PaperBench.

| Category (Proportion) | Rubric Requirement |
| --- | --- |
| Math Fidelity (57.9%) | The code for doing SVD decomposition on MLP.vToxic has been implemented. |
| Algorithmic Logic (42.1%) | The code for generating negative toxic examples for each prompt from GPT-2 has been generated. For each prompt, a negative example (toxic) has been obtained by using PPLM and the toxic vector W as the attribute classifier. |

## 4.4 EXPERIMENTS ON SUPERVISORY SIGNAL DESIGN

Over 20 papers in PaperBench, we extract an average of 237 evaluation units from the three-level guides and 895 atomic facts. After filtering, the final paper fingerprint contains an average of 164 criteria. When comparing to rubrics in PaperBench annotated by human experts, we observe that 79% of the leaf nodes are successfully recalled in our fingerprints, with a precision of 60%. The rubric requirements missed by our fingerprints (21%) mainly come from figure-based or externally

Table 3: Ablation study on the principles of fingerprint design. Performance is averaged over all 20 papers.

| Method | $PR_{root}$ (%) | $PR_{leaf}$ (%) |
|---|---|---|
| 1) w/o Comprehensiveness | 55.9 | 53.5 |
| 2) w/o Atomicity | 58.2 | 57.0 |
| RefP2C | 62.6 | 61.0 |

Table 4: Ablation study on the effect of iteration numbers. "$\Delta$" column shows the change in percentage points (pp) from the previous iteration.

| Iteration | $PR_{root}$ (%) | $\Delta$ (pp) | $PR_{leaf}$ (%) | $\Delta$ (pp) |
|---|---|---|---|---|
| 0 | 52.8 | — | 50.4 | — |
| 1 | 55.8 | +3.0 | 53.9 | +3.5 |
| 2 | 58.8 | +3.0 | 58.0 | +4.1 |
| 3 | 60.2 | +1.4 | 59.7 | +1.7 |
| 4 | 62.6 | +2.4 | 61.0 | +1.3 |
| 5 | 61.4 | -1.2 | 60.8 | -0.2 |

provided information not present in the main textual content. The lower precision mainly stems from our thorough extraction of detailed criteria, including foundational definitions and mathematical formulas, which are ignored by PaperBench. Despite these differences, the high recall demonstrates a strong alignment with the expert-curated rubric, validating our extraction method's effectiveness. More details are provided in Appendix B.4.

To further evaluate the necessity of the proposed fingerprint design principles, we conduct ablation studies as follows: 1) Without comprehensiveness. We only use the framework- and configuration-level guides to extract atomic evaluation criteria, in which other configurations scattered throughout this paper are ignored. 2) Without atomicity. We bypass the standardization step, treating units in three-level guides as the evaluation criteria, each of which may contain multiple facts. The results in Table 3 clearly show a significant performance drop in both variants, demonstrating the critical role of our design principles. Notably, omitting the comprehensiveness principle results in a performance decrease of up to 6.7% in $PR_{root}$ (%) and 7.5% $PR_{leaf}$ (%), highlighting that although the paragraph-by-paragraph scanning process is complex, it remains indispensable and effective.

### 4.5 EXPERIMENTS ON REFLECTIVE CODE DEVELOPMENT

Based on the extracted paper fingerprints, we iteratively revise the generated code through verification, planning, and refinement over multiple loops. To further evaluate the effectiveness of our fingerprint design and reflective code development, we conduct ablation studies on numbers of revision iterations: 1) Without revision (0 iteration): directly using the initial implementation. 2) Revision with varying numbers of iterations.

As shown in Table 4, both the root-level pass ratio ($PR_{root}$) and leaf-level pass ratio ($PR_{leaf}$) generally improve over the first four iterations, with the most substantial gains occurring between iterations 0 and 2. At iteration 4, the performance reaches its peak, achieving 62.6% $PR_{root}$ and 61.0% $PR_{leaf}$. However, the fifth iteration results in a slight decline, indicating diminishing returns and possible overthinking (Xiang et al., 2025) or noise accumulation, which is consistent with observations in (Wan et al., 2024). These results suggest that four iterations offer the best trade-off between performance improvement, and we adopt this as the default setting for our main experiments. Paper-level evaluation results are provided in Appendix B.5.

## 5 CONCLUSION

Focusing on the foundational step of paper reproduction, we introduce RefP2C, a novel reflective framework for paper-to-code development, achieved by designing paper fingerprints and integrating them into a reflective development strategy. RefP2C uses the fingerprints as supervisory signals to drive multiple iterative verification and refinement loops, systematically aligning the code with the paper's implementation details. We conduct extensive experiments on the PaperBench Code-Dev benchmark, and RefP2C achieves state-of-the-art performance, correctly replicating complex implementation details and demonstrating the effectiveness of this reflective framework. In future work, we plan to extend RefP2C to later stages of paper reproduction, including execution and result matching, to further broaden its applicability.

REPRODUCIBILITY STATEMENT

The source code for our proposed RefP2C framework is available in the supplementary materials. Our experiments use the publicly available PaperBench benchmark. Implementation and hyperparameter details are listed in Appendix B.1.

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

# A    METHODS

## A.1    ADDITIONAL DETAILS FOR THE SUPERVISORY SIGNAL DESIGN STAGE

This section provides additional details regarding the Supervisory Signal Design stage. The specifics of each component are illustrated in Figure 5, 6, 7, 8, 9. Furthermore, Figure 10 presents a sample of the results extracted after this process.

---

### Prompt for Guide Extraction

You are a meticulous Software Test Engineer with expertise in machine learning. You will be analyzing a research paper to create a checklist of verifiable facts that can be used to test a code implementation for correctness.

**Background and Core Mission**
Our ultimate goal is to verify that a given codebase is a faithful and accurate reproduction of the research paper. To do this, we need to extract all key **code-level guidance details**.

Your mission is to identify and select all sentences containing these details. These are the specific, actionable claims that must be true in the code. Because these will be used for validation, it is crucial that you only select sentences that are **substantive and informative**.

Informative sentences typically describe:
• **Data & Task:** The exact datasets, benchmarks, or tasks (e.g., "The task is node classification on the Cora dataset.").
• **Data Processing:** Specific data splits, normalization, or augmentation methods.
• **Hyperparameters:** Concrete values for settings like learning rate, batch size, or optimizer.
• **Model Architecture:** The model's structure, layers, or components (e.g., "The model uses GCN layers.").
• **Algorithmic Steps:** Specific computational steps, formulas, or logical flows.
• **Loss Function:** The specific loss function used, including any custom components or equations.
• **Evaluation Metrics:** The exact metrics used for assessment.

**What to IGNORE**
Do NOT select sentences that contain only high-level claims, qualitative discussions, future work, citations, or general background information.

**Output Format**
Your response MUST be ONLY a single, valid JSON array of integers. Each integer in the array corresponds to the index number of a sentence you have selected. If no sentences in the paragraph are relevant, return an empty array [].

**Example Turn**
User provides paragraph: [1]: For the Cora node classification task, our GCN-based model was trained for 200 epochs. [2]: This approach is highly effective. [3]: We used the AdamW optimizer with a learning rate of 0.01. [4]: Performance was measured using the Accuracy metric. [5]: Future work could explore other datasets.

Your required output: [1,3,4]

---

Figure 5: Prompt for Guide Extraction in Supervisory Signal Design stage.

702
703
704
705
706
707
708
709
710
711
712
713
714
715
716
717
718
719
720
721
722
723
724
725
726
727
728
729
730
731
732
733
734
735
736
737
738
739
740
741
742
743
744
745
746
747
748
749
750
751
752
753
754
755

---

### Prompt for Standardization into Atomic Criteria. Part 1 of 2

You are an expert technical writer and software engineer with a knack for clear and natural language. You are tasked with creating precise, verifiable implementation requirements from research papers that are easy for humans to read and understand.

**Instructions**:
**1. Decompose**: Analyze the "Summary Fact" to identify and isolate every distinct, verifiable claim. A claim is verifiable if a reviewer can confirm its implementation by directly inspecting the code, configuration files without needing to re-run the entire experiment to observe its effect.

   **A.** First, identify and preserve indivisible units. Your highest priority is to keep self-contained concepts like equations and algorithms as a single, indivisible unit. Do not break them into multiple atomic facts.
   - **Example (Correct Handling of Equations):**
     - **Input Fact:** "The loss function is $L = \lambda L_{CE} + (1 - \lambda)L_{reg}$."
     - **Result:** This must become **one single criterion**. The entire formula is the fact.
     - **You must not** create separate criteria for $L$, $L_{CE}$, and $L_{reg}$.

   **B.** Then, decompose all other claims into atomic facts. For any information that is *not* a self-contained formula or algorithm, break it down into the smallest meaningful units. A single sentence often contains multiple atomic facts.

   You can break it down into atomic facts such as:

   **• Data and Task Specification**: Identify the specific datasets used and the task being performed. (e.g., "The Cora dataset is used," "The task is node classification.")

   **• Data Handling and Preprocessing**: The specific methods for splitting, normalizing, or augmenting data. (e.g., "Data is split 80/10/10," "Inputs are normalized with a specific mean/std.")

   **• Configuration and Hyperparameters**: Extract specific key-value settings. This is a very common type of fact. (e.g., learning_rate: 0.01, optimizer: 'AdamW', dropout: 0.5, epochs: 200).

   **• Model Architecture and Components**: Pinpoint claims about the model's structure or its parts. (e.g., "The model uses GCN layers," "An attention mechanism is included," "Node features are represented by a learnable embedding vector.").

   **• Algorithmic Process & Calculations**: Isolate specific computational steps or references to formulas. **Note**: If the fact describes a complete, self-contained equation or algorithm (like a specific loss function), treat it as a **single, indivisible unit** and write it as a **single criterion**.

   **• Evaluation**: Identify the specific metrics used to assess performance. (e.g., "Model performance is measured by Accuracy," "The F1-score is reported.").

   **• Package Requirements**: Note any specified libraries, packages, or hardware. (e.g., "The implementation requires PyTorch").

**2. Formulate a Verifiable Criterion**: Your ultimate goal is to generate a **verifiable criterion** for each decomposed fact. Think of each criterion as a self-contained test case description that a code reviewer will use. It must clearly state *what* needs to be verified and *where/when* it applies.

   To do this, formulate a clear and precise "criterion" string by naturally weaving together these two essential components:

   **• The Atomic Fact**: The specific, code-level claim to be checked. This is **atomic fact** you identified in Step 1.

   **• The Scope**: The complete context in which the fact is true. This is the "where" or "when". (e.g., "for the text classification task," "during the main experiments," "in the ablation study").

   For maximum readability and impact, structure your sentence to present the core **Fact** as the main subject, with the **Scope** providing the necessary context. Feel free to use one or two fluent sentences to ensure the criterion is both complete and easy for a human to understand.

   For clarity with complex topics, feel free to use one or two fluent sentences.

---

Figure 6: Prompt for Standardization into Atomic Criteria. Part 1 of 2.

## Prompt for Standardization into Atomic Criteria. Part 2 of 2

**3. Output Format**: Your response must be a valid JSON list of objects, each with a "criterion" key. Inside this string, you must embed distinct XML-style tags: use <fact> and </fact> for the Verifiable Fact, and use <scope> and </scope> for the Scope (the context where the fact is true).

Here are two dummy examples of inputs and expected outputs.

**Example 1: Handling Entities and Hyperparameters**

**Input**

Summary Fact: "In our experiments, we use the AdamW optimizer with a learning rate of 1e-4 and a weight decay of 0.01 on dataset Cora. we use the Adam with a learning rate of 2e-4 on dataset Citeseer."

Reference Sentence: "We tuned hyperparameters separately for each dataset. For experiments on Cora, we used the AdamW optimizer with a learning rate of 1e-4 and weight decay of 0.01. For the Citeseer dataset, we found the standard Adam optimizer with a learning rate of 2e-4 yielded better results."

**Your Output**:

```
[
    {
        "criterion": "The <fact>AdamW optimizer</fact> is used to train the
    model <scope>for the dataset Cora</scope>."
    },
    {
        "criterion": "A <fact>learning rate of 0.0001</fact> is applied
    <scope>when using the AdamW optimizer on the Cora dataset</scope>."
    },
    {
        "criterion": "A <fact>weight decay of 0.01</fact> is used
    <scope>when using the AdamW optimizer on the Cora dataset</scope>."
    },
    {
        "criterion": "The <fact>Adam optimizer</fact> is used to train the
    model <scope>for the dataset Citeseer</scope>."
    },
    {
        "criterion": "A <fact>learning rate of 0.0002</fact> is applied
    <scope>when using the Adam optimizer on the Citeseer dataset</scope>."
    }
]
```

**Example 2: Handling Methods, Architecture, and Processes**

**Input**

Summary Fact: "The actor loss for on-policy updates is the PPO clipped objective, defined as $L^{CLIP}(\theta) = \hat{\mathbb{E}}_t[\min(r_t(\theta)\hat{A}_t, \text{clip}(r_t(\theta), 1 - \epsilon, 1 + \epsilon)\hat{A}_t)]$."

Reference Sentence: "For all on-policy updates, we compute the actor loss using the PPO clipped surrogate objective (Schulman et al., 2017): $L^{CLIP}(\theta) = \hat{\mathbb{E}}_t[\min(r_t(\theta)\hat{A}_t, \text{clip}(r_t(\theta), 1 - \epsilon, 1 + \epsilon)\hat{A}_t)]$, where $r_t(\theta)$ is the probability ratio."

**Your Output**:

```
[
    {
        "criterion": "The <fact>actor loss is calculated using the PPO clipped
    objective:
        L^{CLIP}(θ) = Ê_t[min(r_t(θ)Â_t, clip(r_t(θ), 1 − ε, 1 + ε)Â_t)]</fact>
                    <scope>for all on-policy updates</scope>."
    }
]
```

Please respond with ONLY the list.

Figure 7: Prompt for Standardization into Atomic Criteria. Part 2 of 2.

---

## Prompt for Filtering. Part 1 of 2

You are an expert software engineer and QA lead, specializing in creating actionable engineering checklists from academic research. Your primary goal is to select criteria that are directly and unambiguously verifiable by inspecting a project's source code and configuration files. You must prioritize concrete specifications over abstract concepts.

You will be given a numbered list of checklist criteria that have been grouped because they share the same core <fact>.

**Your Task**:
Your goal is to select the minimum number of criteria necessary to represent all distinct and verifiable implementation details from the list.

Follow this exact algorithm:

1. **Step 1: Group Synonymous Criteria**
   First, mentally group together any criteria that are semantically identical in both their fact and scope. This is the most critical step. You must be very strict. For example, the criterion <fact>entropy coefficient is set to 0</fact> <scope>for the Shadow Hand task</scope> is considered identical to <fact>entropy coefficient is set to 0</fact> <scope>in the training hyperparameters for the Shadow Hand tasks</scope>, and they must be treated as one single group.

2. **Step 2: Identify Unique, Verifiable Groups**
   After grouping, you will be left with a set of unique semantic meanings.

3. **Step 3: Select the Best Representative from Each Unique Group**
   From each unique semantic group, select the single best-phrased criterion that represents it. The "best" criterion is the one that is most valuable to a code reviewer, following these priorities:
   1. Directly Verifiable (Most Important): The claim can be confirmed by looking at the code or a config file (e.g., a specific parameter value 'learning_rate: 0.01', a function call, a class name).
   2. Precise and Unambiguous: It contains specific values and clear actions, leaving no room for interpretation.
   3. Complete and Well-Written: It includes both the core fact and its necessary scope in a professional manner.

4. **Step 4: Final Selection Principle**
   Your final list of selected indices must be as short as possible. Only select multiple representatives if they describe truly distinct, non-overlapping, and verifiable requirements. DO NOT select multiple same representatives. In any case, you must select less than six.

**Your response MUST be a single, valid JSON object with the following two keys**:

- "selected_indices": A list of 1-based integer indices of the final items you selected.

- "reason": A brief explanation of your selection logic, justifying your choice based on the principles above.

**Examples**:

**Example 1: Prioritizing Verifiable Detail**
**Input**:
1. The <fact>model architecture</fact> is <scope>inspired by Transformers</scope>.
2. The <fact>model uses 12 layers of Transformer encoders</fact> <scope>in its main architecture</scope>.
3. The <fact>model's design</fact> considers <scope>long-range dependencies</scope>.

**Your Expected Output**:

```
{
    "selected_indices": [2],
    "reason": "Selected only item 2 because it is the most concrete and directly
    verifiable requirement (12 layers). Items 1 and 3 are abstract concepts, not specific
    implementation details."
}
```

Figure 8: Prompt for Filtering in Supervisory Signal Design stage. Part 1 of 2.

---

**Prompt for Filtering. Part 2 of 2**

**Example 2: Multiple Distinct and Verifiable Scopes**

**Input**:
1. A <fact>dropout of 0.5</fact> is applied <scope>during pre-training</scope>.
2. A <fact>dropout of 0.6</fact> is applied <scope>during fine-tuning</scope>.

**Your Expected Output**:

```
{
    "selected_indices": [1, 2],
    "reason": "Selected two representatives as they describe distinct, veri-
fiable dropout
                values for different training phases (pre-training vs. fine-
tuning)."
}
```

Figure 9: Prompt for Filtering in Supervisory Signal Design stage. Part 2 of 2.

---

**Example: Extracted Guide, Original Sentence and Criteria**

**Guide:**
We use a recurrent policy for the AllegroKuka tasks..

**Original Sentence from Paper:**
*"We use a recurrent policy for the AllegroKuka tasks and an MLP policy for the Shadow Hand and Allegro Hand tasks and use PPO to train them."*

**Extracted Atomic Criteria:**
1. A <fact>recurrent policy</fact> is used <scope>for the AllegroKuka tasks</scope>.
2. An <fact>MLP policy</fact> is used <scope>for the Shadow Hand and Allegro Hand tasks</scope>.
3. The <fact>PPO algorithm</fact> is used to train the policies <scope>for all tasks</scope>.

Figure 10: An example result of our Supervisory Signal Design process, showing the guide, the original sentence from the paper, and the extracted atomic criteria.

## A.2 ADDITIONAL DETAILS FOR THE REFLECTIVE CODE DEVELOPMENT STAGE

This section provides additional details regarding the Reflective Code Development stage. Figures 11 and 12 illustrate the initial implementation of our framework. Figure 13 details the verification process, while Figures 14 and 15 depict the revision planning and refinement steps, respectively.

## Prompt for Initial Implementation in Reflective Code Development stage. Part 1 of 2

You are a professional machine learning engineer. You will be provided with an overall workflow summary of a research paper, four detailed sections from the research paper covering Data, Model, Training, and Evaluation and supplementary information for code reproduction.

Your task is to generate a code framework for implementing the methods described in the summary. The code framework should be a Python script containing only the basic structure including function definitions, class definitions and their corresponding docstrings but no actual implementation code in the function or class.

You should meet these requirements when writing the code framework:

1. Define four classes including Data, Model, Trainer and Evaluator to organize the code framework. For each class, provide a formal class-level docstring that clearly describes the classes's input arguments, all its methods and their purposes.

2. Import necessary packages at the beginning of the script. If the script imports multiple libraries with overlapping or distinct roles (e.g. dgl and torch_geometric), add a brief comment on the same line after each import to clarify its specific usage scenario.

3. Define a main() function to organize the top-level workflow.

4. If you include an `if __name__ == "__main__":` block, it must contain a valid function call such as `main()`.

5. DO NOT implement any actual code inside the functions or classes. The output should solely consist of the basic structure: class definitions, function definitions, and their respective docstrings.

Here is the required example format for a class and a function in the code framework:

```python
class Data(BaseClass):
    """
    A detailed description of what this class represents or does.

    Attributes:
    attr1 (type): Description of attr1.
    attr2 (type): Description of attr2.
    """

    def __init__(self, arg1, arg2, ...):
        """
        A detailed description of what this function does.

        Args (optional):
        arg1 (type): Explanation of arg1.
        arg2 (type): Explanation of arg2.
        """
        pass

    def method_name(self, arg1, arg2, ...):
        """
        A detailed description of what this function does.

        Args (optional):
        arg1 (type): Explanation of arg1.
        arg2 (type): Explanation of arg2.

        Returns (optional):
        result1 (type): Description of the returned result1.
        result2 (type): Description of the returned result2.
        """
        pass
```

Remember do NOT implement any actual code inside the functions or classes. Please respond with only the Python code. No explanations or extra text. The Python code should begin with ```python and end with ```.

Figure 11: Prompt for Initial Implementation in Reflective Code Development stage. Part 1 of 2.

Prompt for Initial Implementation in Reflective Code Development stage. Part 2 of 2

You are a professional machine learning engineer, specializing in code reproduction.

You will be provided with a research paper, supplementary information for code reproduction, the extracted YAML configuration for the experiment, and the code framework. The code framework contains function and class definitions, docstrings, and step comments, but no implementation code. Some steps will include an implementation supplement from the paper, marked with "(paper)", but not all steps will have this.

Additionally, in a multi-turn dialogue, you will receive only one target part of the code framework and need to generate its implementation. Your task is to write a fully complete and human-modifiable class or funtion that faithfully implements every aspect of the described methods and tasks based on the provided code framework and detailed extracted information.

**NOTES**:

  • At each dialogue turn, you will be provided with a list of already imported modules. You MUST NOT modify or remove any of these existing imports. Your response should only include any **NEW** imports that are required for the current code implementation and are not present in the provided list. If new imports are needed, place them at the very top of your generated code block.

  • Strictly follow the structure and function names in the provided code framework. Do not add or remove any functions already present in the code framework.

  • Ensure the code is properly formatted. DO NOT include any placeholders, 'TODO' comments, 'dummy implementation' markers, or similar indications of incomplete code. Every part of the code must be fully implemented to reproduce the paper.

Here is an example for output:

```python
import numpy
import torch
from torch import nn

class Data(BaseClass):
    """
    A detailed description of what this class represents or does.

    Attributes:
    attr1 (type): Description of attr1.
    attr2 (type): Description of attr2.
    ...
    """

    def __init__(self, arg1, arg2, ...):
        """
        A detailed description of what this function does.

        Args (optional):
        arg1 (type): Explanation of arg1.
        arg2 (type): Explanation of arg2.
        ... (type): Additional arguments as needed.
        """
        <your implementation>
```

Please respond with only the Python code. No explanations or extra text. The Python code should begin with ```python and end with ```.

Figure 12: Prompt for Initial Implementation in Reflective Code Development stage. Part 2 of 2.

## Prompt for Verification in Reflective Code Development stage

You are judging a candidate's attempt to reproduce a research paper. You will first be given the research paper, which you should read carefully and refer to as the ground truth for what constitutes a correct implementation and reproduction.

Following that, you will be presented with the candidate's submission: a set of files that attempts to reproduce the paper. To judge the submission, we have prepared full fingperprint describing fine-grained criteria for different aspects of the paper.

Your task is to check the submission for ONE specific criterion from this fingerprint. Please present your answer in 3 parts as follows. Be brief and specific in each part.

**Expected Implementation**
Summarize your expectations in 2-3 sentences. Describe what a correct implementation should contain.

**Actual Findings**
Briefly comment on the relevant code parts in 2-3 sentences. Compare the submission to your expectations.

**Verification Result**
Give a score of 0 or 1. Explain your reasoning in 2-3 concise sentences.

**Other notes**:
 • You must always provide a score. If you have any uncertainties, make them clear in your discussion.
 • All the files and logs from the submission attempt have been provided to you. If anything appears to be missing, assume that the attempt failed to produce it (treat it as a failure).
 • Be strict and thorough in grading your resolution criteria, but do not check for things that are outside of your scope.

Figure 13: Prompt for Verification in Reflective Code Development stage.

## Prompt for Revision Planning in Reflective Code Development stage

You are an expert software architect and project lead. Your task is to analyze a code evaluation report and the corresponding code, then create a clear, step-by-step action plan for a developer to follow.

**Context: evaluation feedback**
The following code was evaluated and failed. Here is the detailed report:

{feedback}

**Context: current code project**
Here is the full code for the project that the feedback refers to:

{code}

**Your task:**
Based on the feedback and the current code, create a concise, actionable plan to fix all issues. You MUST structure your plan into two distinct sections: one for the configuration file ('config.yaml') and one for the Python source code files.

**Important output format**:

• First, provide the plan for the configuration file under the heading '### CONFIG_PLAN'. List the changes for 'config.yaml'. If no changes are needed, write "No changes needed for config.yaml".

• Second, provide the plan for all Python files under the heading '### CODE_PLAN'. Group changes by filename, each with its own '## Code: [filename]' sub-heading.

• Do not write the code itself, only the plan.

**Example output**:

### CONFIG_PLAN

1. In the "training" section, decrease the 'learning_rate' to 1e-5.
2. Under 'pde.convection', set 'beta' to 40.

### CODE_PLAN

## Code: model.py
1. In the 'APTAdapter' class, change the default 'scaling_factor' in the constructor from 2.0 to 4.0.

## Code: main.py
1. Add a 'try...except' block around the 'trainer.train()' call.

Figure 14: Prompt for Revision Planning in Reflective Code Development stage.

## Prompt for Refinement in Reflective Code Development stage

You are an expert-level software engineer with a deep understanding of experimental design and reproducibility in scientific research. Your task is to execute a clear, step-by-step plan to fix a multi-file Python project.

**Code quality requirements:**

• The code you write must be elegant, modular, and maintainable, adhering to Google-style guidelines.

• It must strictly align with the paper's methodology, experimental setup, and evaluation metrics.

• COMPLETE CODE: Your code will be part of the entire project, so please implement complete, reliable, reusable code snippets.

• For any settings, ALWAYS SET A DEFAULT VALUE, USE STRONG TYPING, AND EXPLICIT VARIABLES. AVOID circular imports.

• You MUST FOLLOW the "Data structures and interfaces" from the original design. DO NOT CHANGE ANY DESIGN or use non-existent public methods.

• Before using an external variable or module, make sure you import it first.

• Write out EVERY CODE DETAIL. DO NOT LEAVE TODO comments.

• You must use configuration values from 'config.yaml' where applicable and NOT FABRICATE any new ones.

**Important editing style:**

• Your primary goal is to make the minimum necessary changes to the code to address the plan.

• Preserve the existing code structure, comments, and logic that are not related to the plan.

• Think of your task as applying a precise "patch" or "diff" to the code, not as a complete rewrite.

**Your task:**

You will be given the current version of all files in a project and a revision plan. Based on the plan, you must rewrite the necessary files.

**Instructions for your output:**

• You MUST return the complete, revised code for ALL files in the project, even for files that are not changed.

• Each file's content must be inside its own block, starting with '## Code: [filename]' on a new line, followed by a ```python ... ``` block.

Figure 15: Prompt for Refinement in Reflective Code Development stage.

# B EXPERIMENTS

## B.1 IMPLEMENTATION DETAILS

We adopt `Deepseek-V3` (as a trade-off between performance and cost) for both the supervisory signal design stage and the evaluation phase of the reflective code development stage, while using `o3-mini-high` during the initial implementation, revision planning and refinement stages due to its strong reasoning and coding capabilities. When evaluating on PaperBench Code-Dev, we adopt `o3-mini-high` as recommended. Additionally, we employ `all-MiniLM-L6-v2` sentence-transformers to encode and retrieve paper paragraphs and sentences. The verification-planning-refinement stage is conducted for up to $4$ iterations except section 4.5, with early termination if all evaluation criteria are satisfied.

## B.2 DETAILED ANALYSIS ON PERFORMANCE COMPARISON

We select `lca-on-the-line` for this case study because it shows the largest drop in $PR_{leaf}$ on PaperBench Code-Dev. Compared with PaperCoder, there are 53 evaluation criteria that Paper-Coder implements incorrectly while RefP2C implements them correctly, and 88 criteria for which RefP2C fails but PaperCoder succeeds. For these 88 criteria that RefP2C misses, the failures do not stem from missing mathematical formulas or complex algorithms, but rather reflect a fundamental limitation of LLM-as-judge. In our deep analysis, we observe that using a single rubric to judge implementations leads the agent to misclassify semantically equivalent but syntactically different code. For example, our model's modular accessor `model.feature_extractor()` is functionally correct but is marked as incorrect because the rubric expects direct access to `model.fc`. Likewise, although the code correctly loads all five OOD datasets, the validation script is hardcoded to test only one split, causing every related rubric check to fail. These mismatches occur separately across six datasets (five OOD plus one in-distribution) and cascade into repeated failures, accounting for the vast majority of the 88 regressions.

## B.3 CASE STUDY

To show the correctly revised evaluation criteria by RefP2C, we conduct a case study on the `mechanistic-understanding` paper and grouped the 19 corrections into two simple categories: mathematical fidelity and core algorithmic logic.

**Mathematical Fidelity.** Over half of the correctly revised criteria (12 out of 19, or 57.9%) were about getting math right. For example, PaperCoder skipped the needed matrix transpose before SVD (`id: 1a8266f6`) and left out functions for cosine similarity (`id: 9bbf6a62`) or norm difference (`id: cac04bcb`) between parameters. Using clear and atomic criteria pulled straight from the paper's formulas, our framework revised these to match exactly what the authors wrote.

**Core Algorithmic Logic.** The other 8 criteria (42.1%) were about implementing multi-step procedures correctly. PaperCoder, for instance, turned the toxic-vector step into a simple threshold test instead of ranking and picking the top $k$ (`id: bbdb4b01`), didn't generate toxic examples with PPLM (`id: 3c36d4c4`), and missed the logic to find prompts that lead to a given next token (`id: 52557c05`). By checking each step against our paper fingerprint, we were able to fill in the full logic and make the code follow the paper's exact specifications.

## B.4 DETAILED ANALYSIS ON SUPERVISORY SIGNAL DESIGN

To provide insight into the designed fingerprints, we present a quantitative analysis of the number of criteria at each stage. Table 5 shows the statistics of guides and criteria generated per paper across the 20 documents in our dataset.

The results in Table 5 illustrate the "funnel" effect of our pipeline. The initial guide extraction yields a substantial number of raw information units. The standardization step significantly increases this number as complex sentences are decomposed into multiple atomic criteria. Subsequently, the filtering stages effectively reduce this large set, removing redundancy and irrelevant information, respectively. This process results in final supervisory signals (fingerprint) that is both comprehensive in its coverage and manageable in its size.

Table 5: Average number of criteria per paper at each stage of the supervisory signal design.

| Stage | Avg. Criteria |
|---|---|
| **1. Guide Extraction** | **237.6** |
| - Level 1 (Framework) | 41.4 |
| - Level 2 (Configuration) | 18.9 |
| - Level 3 (Exhaustive-level) | 177.3 |
| **2. After Standardization** | **895.8** |
| - Level 1 | 137.4 |
| - Level 2 | 61.8 |
| - Level 3 | 696.5 |
| **3. After Filtering** | **164.6** |

To validate our automatically generated fingerprint, we compared it against the manually created rubrics in PaperBench using an LLM-based analysis. For each fingerprint criterion, we asked the model whether it matched any requirement in the official rubric. Based on the above results, We designed two evaluation metric: recall and precision. Specifically, $\text{Recall} = \frac{\#\text{covered rubric requirements}}{\#\text{rubric requirements}}$ and $\text{Precision} = \frac{\#\text{matching fingerprint criteria}}{\#\text{fingerprint criteria}}$.

The results show an average recall of 79%, indicating that our extraction process recovers most of the rubric's requirements, while the average precision of 60% means our fingerprint also picks up many extra details that the hand-crafted rubrics leave out. Our detailed review finds that the 21% of rubric items not recalled by our method mostly come from information not in the main paper text. For example, some rubric points rely on author-provided supplements or details shown only in figures, which our text-based approach cannot extract. On the other hand, the 40% precision gap is largely due to a mismatch in granularity: our atomic criteria break down facts more finely than the broader, human-written rubric entries. For instance, in the `mechanistic-understanding` paper the expert rubric has a single requirement saying: "The code for fine-tuning GPT2 using DPO has been implemented. The training uses the following hyper-parameters: a learning rate of 1e-6, batch-size of 4..." By contrast, our fingerprint splits this into separate checks like "The learning rate is set to 1e-6 for DPO training." and "A batch size of 4 is used during DPO training.". The LLM-based matcher may not link these multiple criteria back to the one rubric entry, so it underestimates precision even though all checks may be correct.

### B.5 DETAILED ANALYSIS ON REFLECTIVE CODE DEVELOPMENT

The iterative revision process produces clear, sustained improvement in both root level and leaf level PR scores across all 20 test papers (see Tables 6 and 7). In the earliest stages, from iteration 0 (the initial code draft) through iteration 2, we observe the fastest performance gains because our reflective agent quickly finds and corrects the most obvious discrepancies between the generated code and the paper's specifications. From iteration 2 to iteration 4, scores continue to rise but at a more gradual pace as the agent resolves increasingly subtle implementation details. Most papers reach their highest PR values by the fourth iteration, after which further revisions produce smaller improvements in many cases. The iteration count at which peak performance is achieved varies by paper: simpler methods with well structured algorithms often converge by the third iteration, while more complex architectures or protocols in the experiments sometimes benefit from an additional pass. These findings highlight the efficiency of early corrections and underscore the need to balance the total number of iterations against computational cost when applying our framework in practice. In our experiments, based on the average performance across all papers, we set the maximum number of iterations to four.

Table 6: **PR$_{root}$ (%)** of RefP2C for each paper across iteration 0-5.

| Paper Name | Iter 0 | Iter 1 | Iter 2 | Iter 3 | Iter 4 | Iter 5 |
|---|---|---|---|---|---|---|
| adaptive-pruning | 25.4 | 28.3 | 32.6 | 41.2 | 35.5 | 42.8 |
| all-in-one | 64.1 | 64.6 | 64.8 | 64.2 | 59.1 | 58.1 |
| bam | 68.9 | 70.8 | 74.5 | 77.4 | 77.2 | 77.6 |
| bbox | 35.7 | 42.0 | 43.1 | 44.8 | 45.7 | 50.0 |
| bridging-data-gaps | 57.0 | 57.9 | 67.3 | 72.0 | 76.2 | 62.6 |
| fre | 47.4 | 46.1 | 44.6 | 45.9 | 47.9 | 57.2 |
| ftrl | 26.3 | 23.6 | 24.6 | 26.8 | 29.0 | 27.3 |
| lbcs | 63.5 | 64.0 | 65.5 | 66.0 | 68.5 | 65.1 |
| lca-on-the-line | 30.6 | 33.6 | 37.9 | 40.3 | 39.0 | 36.1 |
| mechanistic-understanding | 76.1 | 86.3 | 91.9 | 93.7 | 91.9 | 91.9 |
| pinn | 49.4 | 55.9 | 54.4 | 41.8 | 62.3 | 47.5 |
| rice | 50.0 | 45.8 | 47.7 | 49.2 | 63.7 | 58.6 |
| robust-clip | 30.2 | 38.5 | 41.3 | 41.2 | 42.3 | 43.0 |
| sample-specific-masks | 64.5 | 69.1 | 73.8 | 72.9 | 74.3 | 75.2 |
| sapg | 64.3 | 70.0 | 74.9 | 75.0 | 74.2 | 74.6 |
| sequential-neural-score-estimation | 62.3 | 63.9 | 68.5 | 74.6 | 78.9 | 78.9 |
| stay-on-topic-with-classifier-free-guidance | 54.0 | 62.9 | 64.5 | 61.0 | 66.8 | 68.7 |
| stochastic-interpolants | 74.7 | 71.5 | 72.9 | 80.7 | 88.2 | 87.2 |
| test-time-model-adaptation | 61.8 | 67.3 | 76.0 | 76.3 | 76.9 | 77.2 |
| what-will-my-model-forget | 49.7 | 53.7 | 55.8 | 58.0 | 53.7 | 49.3 |
| **Average** | 52.8 | 55.8 | 58.8 | 60.2 | 62.6 | 61.4 |

Table 7: **PR$_{leaf}$ (%)** of RefP2C for each paper across iteration 0-5.

| Paper Name | Iter 0 | Iter 1 | Iter 2 | Iter 3 | Iter 4 | Iter 5 |
|---|---|---|---|---|---|---|
| adaptive-pruning | 34.9 | 44.2 | 48.8 | 58.1 | 52.3 | 59.3 |
| all-in-one | 44.6 | 46.7 | 44.6 | 45.7 | 52.2 | 47.8 |
| bam | 56.1 | 56.5 | 56.9 | 58.8 | 57.6 | 57.6 |
| bbox | 37.9 | 38.6 | 41.4 | 44.1 | 38.6 | 41.4 |
| bridging-data-gaps | 54.5 | 56.4 | 70.9 | 72.7 | 80.0 | 61.8 |
| fre | 48.4 | 51.3 | 56.2 | 60.5 | 59.8 | 51.5 |
| ftrl | 36.7 | 35.8 | 36.7 | 39.2 | 37.5 | 37.5 |
| lbcs | 37.1 | 41.5 | 46.3 | 49.5 | 50.3 | 51.6 |
| lca-on-the-line | 19.6 | 21.7 | 22.9 | 24.3 | 21.1 | 24.9 |
| mechanistic-understanding | 69.4 | 80.6 | 83.3 | 86.1 | 83.3 | 83.3 |
| pinn | 89.7 | 92.1 | 89.7 | 80.2 | 92.1 | 91.3 |
| rice | 60.1 | 53.9 | 51.7 | 57.9 | 66.9 | 68.0 |
| robust-clip | 38.6 | 44.3 | 48.6 | 48.6 | 45.7 | 47.1 |
| sample-specific-masks | 67.8 | 69.0 | 77.0 | 75.9 | 75.9 | 79.3 |
| sapg | 50.6 | 70.1 | 75.3 | 76.6 | 77.9 | 76.6 |
| sequential-neural-score-estimation | 56.7 | 55.2 | 65.7 | 67.2 | 70.1 | 71.6 |
| stay-on-topic-with-classifier-free-guidance | 48.6 | 51.4 | 54.3 | 52.9 | 57.1 | 61.4 |
| stochastic-interpolants | 63.8 | 62.1 | 67.2 | 67.2 | 75.9 | 75.9 |
| test-time-model-adaptation | 51.2 | 60.5 | 75.6 | 76.7 | 75.6 | 75.6 |
| what-will-my-model-forget | 41.4 | 45.7 | 47.6 | 51.9 | 50.1 | 51.4 |
| **Average** | 50.4 | 53.9 | 58.0 | 59.7 | 61.0 | 60.8 |

### B.6 COST ANALYSIS

We further report the financial costs associated with fingerprint design and reflective code development. As shown in Figure 16, the majority of the cost comes from fingerprint design ($ 3.95) and a single code refinement loop ($ 1.85). When implementing the initial code without any verification or refinement, the framework only includes extraction at the framework and configuration levels and the initial implementation stage. In this scenario, compared with PaperCoder (Seo et al., 2025), RefP2C achieves both a lower cost ($ 0.63 vs. $ 0.69) and higher performance (52.8% vs. 45.1%, Table 4), further demonstrating its effectiveness.

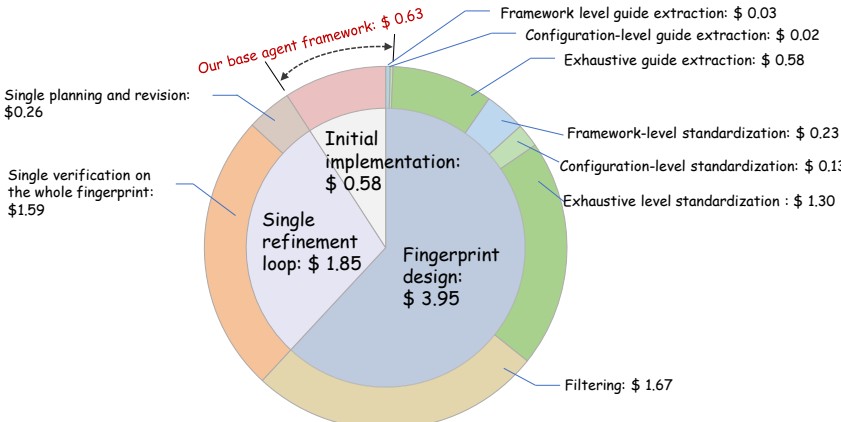

Figure 16: The cost breakdown for an average run of the framework per paper.

## THE USE OF LARGE LANGUAGE MODELS (LLMS)

Throughout the preparation of this manuscript, we utilized large language models (LLMs) primarily to polish the writing. Their use was limited to improving clarity, conciseness, and correcting grammatical errors. The core research ideas, methodology, and scientific conclusions were conceived and articulated entirely by the authors.

