# OpenReview forum: "RefP2C: Reflective Paper-to-Code Development Enabled by Fine-Grained Verification"
_ICLR.cc/2026/Conference — ICLR 2026 Conference Withdrawn Submission_

### Official Review · Reviewer_k9gt · 2025-10-31

**Soundness:** 2
**Presentation:** 3
**Contribution:** 2
**Rating:** 2
**Confidence:** 4

**Summary:**

The paper proposes RefP2C, a reflective agent framework for paper-to-code development, targeting the challenging task of reproducing machine learning research papers without access to original implementation. The core idea involves extracting fine-grained, atomic “fingerprint” criteria from the paper—serving as pass/fail supervisory signals—and driving iterative code development, verification, and revision guided by these criteria. Experiments on the PaperBench Code-Dev benchmark indicate that RefP2C improves code fidelity to the original paper over leading baselines by a significant margin (notably +13% root-level pass ratio), and provides some ablation and case analyses to support the effectiveness of both fingerprinting and iterative refinement. However, the style of the paper is more like a **project technical report**, other than an academic paper.

**Strengths:**

**Clear Motivation**: The work is well-motivated with a thorough diagnosis of pain points in automated paper reproduction, explicitly targeting the code development phase and leveraging human-inspired checklist methodologies as a guiding insight.

**Technical Soundness**: The framework is described with clarity and attention to detail, including explicit breakdowns of the multi-stage pipeline (guide extraction, standardization to atomic criteria, filtering, and reflective development). The iterative verification and refinement loop is well-articulated and grounded in the human analogy.

**Strong Empirical Results**: Table 1 demonstrates substantial improvements (+13 percentage points) in code replication accuracy ($\mathbf{PR}{\text{root}}$ and $\mathbf{PR}{\text{leaf}}$) over state-of-the-art baselines, with extensive per-paper and per-category breakdowns (see Figure 4 and corresponding analysis).

**Methodological Transparency**: The appendices and explanations give concrete examples and promptsfor each pipeline step, supporting reproducibility and reader comprehension of the core procedures.

**Weaknesses:**

**Engineering Contribution, Not Scientific Advancement**: This work is best described as a complex engineering system that orchestrates multiple LLM agents with prompt engineering and heuristics. While the system may be useful in practice, it does not offer new algorithms, models, or theoretical frameworks. The contribution is incremental and application-specific.

**Fingerprint Overlap with Human Rubric Undermines Claimed Generalization**:
- While RefP2C does not directly access the evaluation rubric, we observe that its automatically extracted fingerprint achieves 79% recall against human-authored leaf criteria (rubric requirements). (Line 1316, Appendix B.4) This high overlap raises concerns that the observed performance gains are not due to the framework’s superior reasoning or code generation capabilities, but rather to the fact that **the extracted fingerprint implicitly “covers” the evaluation criteria**.
- In other words, the system benefits from a statistical alignment between what it extracts and what it is later evaluated against. This is particularly problematic given that the fingerprint is derived from the same textual source (the paper) that human rubric designers also rely on. As the authors note at Line 1318, “the 21% of rubric items not recalled by our method mainly come from figure-based or externally provided information not present in the main textual content,” which further confirms that the fingerprint is essentially a textual summary of the rubric, albeit constructed independently.
- Consequently, the leaf-level PR improvements may not reflect a more capable code generation process, but rather a better-informed one—i.e., the model is prompted with **implementation details that happen to match the evaluation checklist**.

**Intuition-Driven Design Without Principle or Justification**:
- The paper introduces a multi-stage pipeline (guide extraction → fingerprint standardisation → verification → planning → refinement) that is rich in terminology and details but poor in principle. Each procedure’s role is described operationally, yet why this particular decomposition is necessary, is never established. And there are many details that do not have clear definitions and can not be verified. Here are some examples:
	- Fingerprint atomicity is asserted, not justified: Section 3.1 declares that criteria must be “atomic” so that a “clear pass/fail judgment” is possible. However, atomicity is defined syntactically (“one fact-scope pair”) rather than functionally.
No evidence is given that this particular grain size maximizes information per criterion or minimizes verifier error. A coarser or finer split might reduce false positives; the paper is silent on this trade-off.
	- Violation of the Principle of Parsimony: For example, in Line 298, the manuscript introduces “short-term” and “long-term memory” without defining their operational difference beyond ordinary context buffering. No ablation shows that explicitly labeling prior edits as “memory” improves performance over a simple concatenation of previous code and feedback. The design philosophy in this paper runs counter to the principle of parsimony (Occam’s Razor): entities should not be multiplied beyond necessity.
	- Iteration cap is arbitrary: The system stops after four refinement rounds because “performance peaks on average” (Table 4). This ceiling is data-driven on the same 20 papers used for reporting, inviting over-fitting. No convergence criterion (e.g., successive entropy of failure labels, stability of planned edits) is provided.
- In short, the architecture is intuitively reasonable but scientifically unjustified. Without ablations that isolate the marginal utility of each procedure, without sensitivity analyses on prompt design, and without convergence guarantees, the reader is asked to accept an ad-hoc workflow whose complexity may obscure rather than enable robust paper-to-code reproduction.

**The Claimed “Failure of Prior Work” Is Not Substantiated**:
- In the abstract, the authors assert that
“current paper reproduction methods fail to effectively adopt [reflection with explicit feedback] … mainly arising from diverse paper patterns, complex method modules, and varied configurations.”
This statement is repeated in Section 1, yet no concrete prior method is cited as having tried and failed to provide explicit feedback.

**Questions:**

- Where is the proof that earlier methods actually tried and failed to use “explicit feedback” during reflection, or is the stated gap just an assumption?
- How do you design the whole system? Is this system built solely on experience and intuition?
- Can you provide more detailed ablations? For example, did you test whether keeping or dropping the so-called “short-term” and “long-term memory” changes the results at all?
- How do you know the 79 % overlap between your fingerprint and the alignment with human rubric is not the real reason for your higher scores?

---

### Official Review · Reviewer_De1w · 2025-11-01

**Soundness:** 2
**Presentation:** 4
**Contribution:** 3
**Rating:** 4
**Confidence:** 4

**Summary:**

This paper proposes RefP2C, a reflective multi-agent framework to automate paper-to-code development. The core idea is to automatically extract a paper's "fingerprint"—a set of atomic, verifiable criteria—to serve as a checklist. This fingerprint guides an iterative loop of verification, planning, and refinement, where agents correct the code to match the paper's specifications. Experiments on the PaperBench Code-Dev benchmark show significant performance improvements over strong baselines.

**Strengths:**

1. The paper is well-written and addresses the critical and foundational problem of implementation fidelity in paper-to-code reproduction.
2. It proposes a novel and human-inspired framework (RefP2C) that uses an automatically extracted "fingerprint" as a verifiable supervisory signal for reflection.
3. The empirical validation is strong, showing significant SOTA gains over strong baselines on the PaperBench Code-Dev benchmark.

**Weaknesses:**

**Major**

1. My major concern is that the evaluation relies entirely on a single model, o3-mini-high, for its main experiments (with Deepseek-V3 used for some sub-tasks). This makes it difficult to distinguish the framework's effectiveness from the specific capabilities of this particular model. It is unclear how RefP2C would perform with other, potentially stronger or architecturally different models (e.g., Claude 4.5 or Gemini-2.5).
2. The computational cost-benefit of the core "reflection" mechanism is not fully analyzed. The "Initial Implementation" (Iteration 0) already achieves 52.8% $PR_{root}$, outperforming all baselines. The subsequent four iterations provide only a +9.8% relative gain (to 62.6%) while adding significant cost. It is unclear if this trade-off is practical.
3. The paper demonstrates a potential contradictory stance on LLM-as-judge. It relies on its internal "verifier" agent to be accurate, yet (in Appendix B.2) blames the *external* benchmark judge for being "rigid" and mis-evaluating semantically equivalent code. This raises questions about the reliability of both the internal loop and the final reported scores.

**Minor:**

1. There is a mismatch between the agent's optimization objective (the auto-generated "fingerprint") and the final evaluation target (the human-curated "rubric"). The paper reports only 79% recall for its fingerprint against the rubric, meaning the framework cannot optimize for 21% of the criteria it is ultimately judged on.

**Questions:**

1. Could you quantify the computational overhead (e.g., total tokens, API cost in USD, wall-clock time) required to run the full 4-iteration pipeline for an average paper, versus the Iteration 0 baseline?
2. Following up on the reliance on o3-mini-high: How does the framework's performance (both $PR_{root}$ and the quality of the *fingerprint itself*) change when a different model family (e.g., Claude 4.5 or Gemini-2.5) is used?
3. Given that 21% of rubric items are missed (often from figures), how do you see the framework handling figure-heavy papers (e.g., in computer vision or systems)? Are you exploring multi-modal inputs for the fingerprint extractor?

---

### Official Review · Reviewer_VQSn · 2025-11-07

**Soundness:** 3
**Presentation:** 3
**Contribution:** 2
**Rating:** 2
**Confidence:** 4

**Summary:**

This paper introduces RefP2C, a framework for automatically reproducing machine learning research papers by generating code implementations. The key innovation is the extraction of a paper's "fingerprint" - a comprehensive set of atomic, verifiable criteria that serve as supervisory signals for iterative code refinement. The framework consists of two stages: (1) extracting fine-grained evaluation criteria from papers, and (2) using these criteria to guide reflective code development through verification and refinement loops.

**Strengths:**

1. Strong Empirical Results: The 13.0% absolute $PR_{root}$ improvement over the next-best baseline on a standard benchmark is significant.
2. The paper is well written and easy following.

**Weaknesses:**

1. Limited Benchmark Generalizability: The paper's evaluation is confined solely to the PaperBench Code-Dev benchmark. This exclusivity makes it unclear how the RefP2C framework would perform against other relevant benchmarks, such as SciReplicate-bench [1] or Paper2Code Bench [2], which might assess different aspects of reproduction or use different evaluation methods.

2. Limited Model Diversity: While the framework uses Deepseek-V3 for fingerprint design, all core coding and reasoning components (initial implementation, revision planning, and refinement) are evaluated using only O3-mini-high. This makes it difficult to determine if the framework's success is generalizable or tied to the specific capabilities of O3-mini-high, leaving its performance with other large language models unknown.


3. Flawed Evaluation Methodology and Lack of Execution: The paper's evaluation relies on the PaperBench Code-Dev benchmark, which uses an "LLM-as-judge". It never verifies if the generated code can actually be executed or if it reproduces the original paper's metrics. The paper explicitly defers "execution and result matching" to future work, which is a significant gap, as the ultimate goal of successful reproduction remains unverified.


[1] scireplicate-bench: benchmarking llms in agent-driven algorithmic reproduction from research papers.

[2] Paper2Code: Automating Code Generation from Scientific Papers in Machine Learning.

**Questions:**

1. Incomplete error analysis: The paper doesn't provide sufficient analysis of failure modes. What types of implementations consistently fail?

2. How does your framework handle cases where the original paper lacks certain information or implementation details?

---

### Note · Authors · 2025-12-04

I have read and agree with the venue's withdrawal policy on behalf of myself and my co-authors.